# Convolutional Neural Networks on Graphs with Fast Localized Spectral Filtering

**Michaël Defferrard**       **Xavier Bresson**       **Pierre Vandergheynst**

EPFL, Lausanne, Switzerland
{michael.defferrard,xavier.bresson,pierre.vandergheynst}@epfl.ch

## Abstract

In this work, we are interested in generalizing convolutional neural networks (CNNs) from low-dimensional regular grids, where image, video and speech are represented, to high-dimensional irregular domains, such as social networks, brain connectomes or words' embedding, represented by graphs. We present a formulation of CNNs in the context of spectral graph theory, which provides the necessary mathematical background and efficient numerical schemes to design fast localized convolutional filters on graphs. Importantly, the proposed technique offers the same linear computational complexity and constant learning complexity as classical CNNs, while being universal to any graph structure. Experiments on MNIST and 20NEWS demonstrate the ability of this novel deep learning system to learn local, stationary, and compositional features on graphs.

## 1 Introduction

Convolutional neural networks [19] offer an efficient architecture to extract highly meaningful statistical patterns in large-scale and high-dimensional datasets. The ability of CNNs to learn local stationary structures and compose them to form multi-scale hierarchical patterns has led to breakthroughs in image, video, and sound recognition tasks [18]. Precisely, CNNs extract the local stationarity property of the input data or signals by revealing local features that are shared across the data domain. These similar features are identified with localized convolutional filters or kernels, which are learned from the data. Convolutional filters are shift- or translation-invariant filters, meaning they are able to recognize identical features independently of their spatial locations. Localized kernels or compactly supported filters refer to filters that extract local features independently of the input data size, with a support size that can be much smaller than the input size.

User data on social networks, gene data on biological regulatory networks, log data on telecommunication networks, or text documents on word embeddings are important examples of data lying on irregular or non-Euclidean domains that can be structured with graphs, which are universal representations of heterogeneous pairwise relationships. Graphs can encode complex geometric structures and can be studied with strong mathematical tools such as spectral graph theory [6].

A generalization of CNNs to graphs is not straightforward as the convolution and pooling operators are only defined for regular grids. This makes this extension challenging, both theoretically and implementation-wise. The major bottleneck of generalizing CNNs to graphs, and one of the primary goals of this work, is the definition of localized graph filters which are efficient to evaluate and learn. Precisely, the main contributions of this work are summarized below.

1. **Spectral formulation.** A spectral graph theoretical formulation of CNNs on graphs built on established tools in graph signal processing (GSP). [31].

2. **Strictly localized filters.** Enhancing [4], the proposed spectral filters are provable to be strictly localized in a ball of radius $K$, i.e. $K$ hops from the central vertex.

3. **Low computational complexity.** The evaluation complexity of our filters is linear w.r.t. the filters support's size $K$ and the number of edges $|\mathcal{E}|$. Importantly, as most real-world graphs are highly sparse, we have $|\mathcal{E}| \ll n^2$ and $|\mathcal{E}| = kn$ for the widespread $k$-nearest neighbor

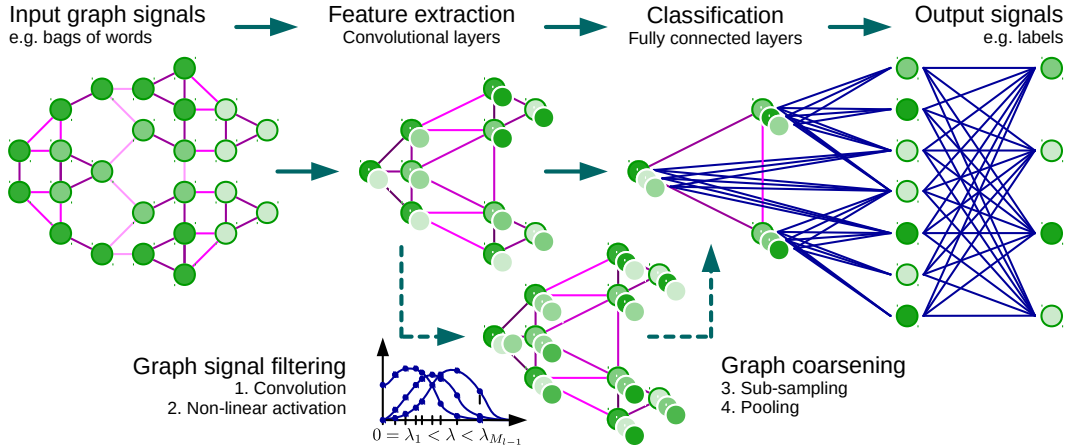

Figure 1: Architecture of a CNN on graphs and the four ingredients of a (graph) convolutional layer.

(NN) graphs, leading to a linear complexity w.r.t the input data size $n$. Moreover, this method avoids the Fourier basis altogether, thus the expensive eigenvalue decomposition (EVD) necessary to compute it as well as the need to store the basis, a matrix of size $n^2$. That is especially relevant when working with limited GPU memory. Besides the data, our method only requires to store the Laplacian, a sparse matrix of $|\mathcal{E}|$ non-zero values.

4. **Efficient pooling.** We propose an efficient pooling strategy on graphs which, after a rearrangement of the vertices as a binary tree structure, is analog to pooling of 1D signals.

5. **Experimental results.** We present multiple experiments that ultimately show that our formulation is (i) a useful model, (ii) computationally efficient and (iii) superior both in accuracy and complexity to the pioneer spectral graph CNN introduced in [4]. We also show that our graph formulation performs similarly to a classical CNNs on MNIST and study the impact of various graph constructions on performance. The TensorFlow [1] code to reproduce our results and apply the model to other data is available as an open-source software.[1]

## 2    Proposed Technique

Generalizing CNNs to graphs requires three fundamental steps: (i) the design of localized convolutional filters on graphs, (ii) a graph coarsening procedure that groups together similar vertices and (iii) a graph pooling operation that trades spatial resolution for higher filter resolution.

### 2.1    Learning Fast Localized Spectral Filters

There are two strategies to define convolutional filters; either from a spatial approach or from a spectral approach. By construction, spatial approaches provide filter localization via the finite size of the kernel. However, although graph convolution in the spatial domain is conceivable, it faces the challenge of matching local neighborhoods, as pointed out in [4]. Consequently, there is no unique mathematical definition of translation on graphs from a spatial perspective. On the other side, a spectral approach provides a well-defined localization operator on graphs via convolutions with a Kronecker delta implemented in the spectral domain [31]. The convolution theorem [22] defines convolutions as linear operators that diagonalize in the Fourier basis (represented by the eigenvectors of the Laplacian operator). However, a filter defined in the spectral domain is not naturally localized and translations are costly due to the $\mathcal{O}(n^2)$ multiplication with the graph Fourier basis. Both limitations can however be overcome with a special choice of filter parametrization.

**Graph Fourier Transform.**    We are interested in processing signals defined on undirected and connected graphs $\mathcal{G} = (\mathcal{V}, \mathcal{E}, W)$, where $\mathcal{V}$ is a finite set of $|\mathcal{V}| = n$ vertices, $\mathcal{E}$ is a set of edges and $W \in \mathbb{R}^{n \times n}$ is a weighted adjacency matrix encoding the connection weight between two vertices. A signal $x : \mathcal{V} \to \mathbb{R}$ defined on the nodes of the graph may be regarded as a vector $x \in \mathbb{R}^n$ where $x_i$ is the value of $x$ at the $i^{th}$ node. An essential operator in spectral graph analysis is the graph Laplacian [6], which combinatorial definition is $L = D - W \in \mathbb{R}^{n \times n}$ where $D \in \mathbb{R}^{n \times n}$ is the

diagonal degree matrix with $D_{ii} = \sum_j W_{ij}$, and normalized definition is $L = I_n - D^{-1/2}WD^{-1/2}$ where $I_n$ is the identity matrix. As $L$ is a real symmetric positive semidefinite matrix, it has a complete set of orthonormal eigenvectors $\{u_l\}_{l=0}^{n-1} \in \mathbb{R}^n$, known as the graph Fourier modes, and their associated ordered real nonnegative eigenvalues $\{\lambda_l\}_{l=0}^{n-1}$, identified as the frequencies of the graph. The Laplacian is indeed diagonalized by the Fourier basis $U = [u_0, \dots, u_{n-1}] \in \mathbb{R}^{n \times n}$ such that $L = U\Lambda U^T$ where $\Lambda = \mathrm{diag}([\lambda_0, \dots, \lambda_{n-1}]) \in \mathbb{R}^{n \times n}$. The graph Fourier transform of a signal $x \in \mathbb{R}^n$ is then defined as $\hat{x} = U^T x \in \mathbb{R}^n$, and its inverse as $x = U\hat{x}$ [31]. As on Euclidean spaces, that transform enables the formulation of fundamental operations such as filtering.

**Spectral filtering of graph signals.** As we cannot express a meaningful translation operator in the vertex domain, the convolution operator on graph $*_\mathcal{G}$ is defined in the Fourier domain such that $x *_\mathcal{G} y = U((U^T x) \odot (U^T y))$, where $\odot$ is the element-wise Hadamard product. It follows that a signal $x$ is filtered by $g_\theta$ as

$$y = g_\theta(L)x = g_\theta(U\Lambda U^T)x = Ug_\theta(\Lambda)U^T x. \tag{1}$$

A non-parametric filter, i.e. a filter whose parameters are all free, would be defined as

$$g_\theta(\Lambda) = \mathrm{diag}(\theta), \tag{2}$$

where the parameter $\theta \in \mathbb{R}^n$ is a vector of Fourier coefficients.

**Polynomial parametrization for localized filters.** There are however two limitations with non-parametric filters: (i) they are not localized in space and (ii) their learning complexity is in $\mathcal{O}(n)$, the dimensionality of the data. These issues can be overcome with the use of a polynomial filter

$$g_\theta(\Lambda) = \sum_{k=0}^{K-1} \theta_k \Lambda^k, \tag{3}$$

where the parameter $\theta \in \mathbb{R}^K$ is a vector of polynomial coefficients. The value at vertex $j$ of the filter $g_\theta$ centered at vertex $i$ is given by $(g_\theta(L)\delta_i)_j = (g_\theta(L))_{i,j} = \sum_k \theta_k (L^k)_{i,j}$, where the kernel is localized via a convolution with a Kronecker delta function $\delta_i \in \mathbb{R}^n$. By [12, Lemma 5.2], $d_\mathcal{G}(i,j) > K$ implies $(L^K)_{i,j} = 0$, where $d_\mathcal{G}$ is the shortest path distance, i.e. the minimum number of edges connecting two vertices on the graph. Consequently, spectral filters represented by $K^{\text{th}}$-order polynomials of the Laplacian are exactly $K$-localized. Besides, their learning complexity is $\mathcal{O}(K)$, the support size of the filter, and thus the same complexity as classical CNNs.

**Recursive formulation for fast filtering.** While we have shown how to learn localized filters with $K$ parameters, the cost to filter a signal $x$ as $y = Ug_\theta(\Lambda)U^T x$ is still high with $\mathcal{O}(n^2)$ operations because of the multiplication with the Fourier basis $U$. A solution to this problem is to parametrize $g_\theta(L)$ as a polynomial function that can be computed recursively from $L$, as $K$ multiplications by a sparse $L$ costs $\mathcal{O}(K|\mathcal{E}|) \ll \mathcal{O}(n^2)$. One such polynomial, traditionally used in GSP to approximate kernels (like wavelets), is the Chebyshev expansion [12]. Another option, the Lanczos algorithm [33], which constructs an orthonormal basis of the Krylov subspace $\mathcal{K}_K(L,x) = \mathrm{span}\{x, Lx, \dots, L^{K-1}x\}$, seems attractive because of the coefficients' independence. It is however more convoluted and thus left as a future work.

Recall that the Chebyshev polynomial $T_k(x)$ of order $k$ may be computed by the stable recurrence relation $T_k(x) = 2xT_{k-1}(x) - T_{k-2}(x)$ with $T_0 = 1$ and $T_1 = x$. These polynomials form an orthogonal basis for $L^2([-1,1], dy/\sqrt{1-y^2})$, the Hilbert space of square integrable functions with respect to the measure $dy/\sqrt{1-y^2}$. A filter can thus be parametrized as the truncated expansion

$$g_\theta(\Lambda) = \sum_{k=0}^{K-1} \theta_k T_k(\tilde{\Lambda}), \tag{4}$$

of order $K-1$, where the parameter $\theta \in \mathbb{R}^K$ is a vector of Chebyshev coefficients and $T_k(\tilde{\Lambda}) \in \mathbb{R}^{n \times n}$ is the Chebyshev polynomial of order $k$ evaluated at $\tilde{\Lambda} = 2\Lambda/\lambda_{max} - I_n$, a diagonal matrix of scaled eigenvalues that lie in $[-1, 1]$. The filtering operation can then be written as $y = g_\theta(L)x = \sum_{k=0}^{K-1} \theta_k T_k(\tilde{L})x$, where $T_k(\tilde{L}) \in \mathbb{R}^{n \times n}$ is the Chebyshev polynomial of order $k$ evaluated at the scaled Laplacian $\tilde{L} = 2L/\lambda_{max} - I_n$. Denoting $\bar{x}_k = T_k(\tilde{L})x \in \mathbb{R}^n$, we can use the recurrence relation to compute $\bar{x}_k = 2\tilde{L}\bar{x}_{k-1} - \bar{x}_{k-2}$ with $\bar{x}_0 = x$ and $\bar{x}_1 = \tilde{L}x$. The entire filtering operation $y = g_\theta(L)x = [\bar{x}_0, \dots, \bar{x}_{K-1}]\theta$ then costs $\mathcal{O}(K|\mathcal{E}|)$ operations.

**Learning filters.**   The $j^{\text{th}}$ output feature map of the sample $s$ is given by

$$y_{s,j} = \sum_{i=1}^{F_{in}} g_{\theta_{i,j}}(L)x_{s,i} \in \mathbb{R}^n, \tag{5}$$

where the $x_{s,i}$ are the input feature maps and the $F_{in} \times F_{out}$ vectors of Chebyshev coefficients $\theta_{i,j} \in \mathbb{R}^K$ are the layer's trainable parameters. When training multiple convolutional layers with the backpropagation algorithm, one needs the two gradients

$$\frac{\partial E}{\partial \theta_{i,j}} = \sum_{s=1}^{S} [\bar{x}_{s,i,0}, \dots, \bar{x}_{s,i,K-1}]^T \frac{\partial E}{\partial y_{s,j}} \qquad \text{and} \qquad \frac{\partial E}{\partial x_{s,i}} = \sum_{j=1}^{F_{out}} g_{\theta_{i,j}}(L)\frac{\partial E}{\partial y_{s,j}}, \tag{6}$$

where $E$ is the loss energy over a mini-batch of $S$ samples. Each of the above three computations boils down to $K$ sparse matrix-vector multiplications and one dense matrix-vector multiplication for a cost of $\mathcal{O}(K|\mathcal{E}|F_{in}F_{out}S)$ operations. These can be efficiently computed on parallel architectures by leveraging tensor operations. Eventually, $[\bar{x}_{s,i,0}, \dots, \bar{x}_{s,i,K-1}]$ only needs to be computed once.

## 2.2   Graph Coarsening

The pooling operation requires meaningful neighborhoods on graphs, where similar vertices are clustered together. Doing this for multiple layers is equivalent to a multi-scale clustering of the graph that preserves local geometric structures. It is however known that graph clustering is NP-hard [5] and that approximations must be used. While there exist many clustering techniques, e.g. the popular spectral clustering [21], we are most interested in multilevel clustering algorithms where each level produces a coarser graph which corresponds to the data domain seen at a different resolution. Moreover, clustering techniques that reduce the size of the graph by a factor two at each level offers a precise control on the coarsening and pooling size. In this work, we make use of the coarsening phase of the Graclus multilevel clustering algorithm [9], which has been shown to be extremely efficient at clustering a large variety of graphs. Algebraic multigrid techniques on graphs [28] and the Kron reduction [32] are two methods worth exploring in future works.

Graclus [9], built on Metis [16], uses a greedy algorithm to compute successive coarser versions of a given graph and is able to minimize several popular spectral clustering objectives, from which we chose the normalized cut [30]. Graclus' greedy rule consists, at each coarsening level, in picking an unmarked vertex $i$ and matching it with one of its unmarked neighbors $j$ that maximizes the local normalized cut $W_{ij}(1/d_i + 1/d_j)$. The two matched vertices are then marked and the coarsened weights are set as the sum of their weights. The matching is repeated until all nodes have been explored. This is an very fast coarsening scheme which divides the number of nodes by approximately two (there may exist a few singletons, non-matched nodes) from one level to the next coarser level.

## 2.3   Fast Pooling of Graph Signals

Pooling operations are carried out many times and must be efficient. After coarsening, the vertices of the input graph and its coarsened versions are not arranged in any meaningful way. Hence, a direct application of the pooling operation would need a table to store all matched vertices. That would result in a memory inefficient, slow, and hardly parallelizable implementation. It is however possible to arrange the vertices such that a graph pooling operation becomes as efficient as a 1D pooling. We proceed in two steps: (i) create a balanced binary tree and (ii) rearrange the vertices. After coarsening, each node has either two children, if it was matched at the finer level, or one, if it was not, i.e the node was a singleton. From the coarsest to finest level, fake nodes, i.e. disconnected nodes, are added to pair with the singletons such that each node has two children. This structure is a balanced binary tree: (i) regular nodes (and singletons) either have two regular nodes (e.g. level 1 vertex 0 in Figure 2) or (ii) one singleton and a fake node as children (e.g. level 2 vertex 0), and (iii) fake nodes always have two fake nodes as children (e.g. level 1 vertex 1). Input signals are initialized with a neutral value at the fake nodes, e.g. 0 when using a ReLU activation with max pooling. Because these nodes are disconnected, filtering does not impact the initial neutral value. While those fake nodes do artificially increase the dimensionality thus the computational cost, we found that, in practice, the number of singletons left by Graclus is quite low. Arbitrarily ordering the nodes at the coarsest level, then propagating this ordering to the finest levels, i.e. node $k$ has nodes $2k$ and $2k + 1$ as children, produces a regular ordering in the finest level. Regular in the sense that adjacent nodes are hierarchically merged at coarser levels. Pooling such a rearranged graph signal is

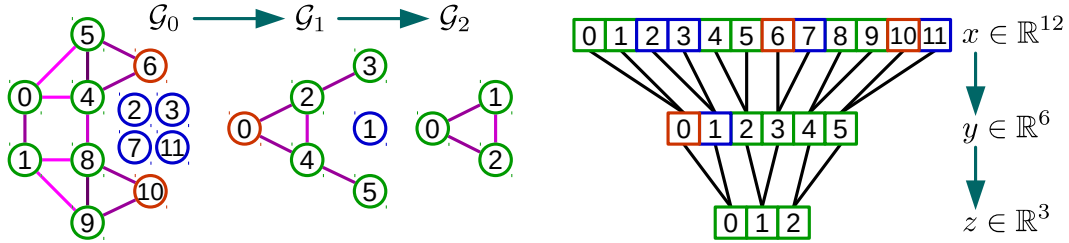

Figure 2: **Example of Graph Coarsening and Pooling.** Let us carry out a max pooling of size 4 (or two poolings of size 2) on a signal $x \in \mathbb{R}^8$ living on $\mathcal{G}_0$, the finest graph given as input. Note that it originally possesses $n_0 = |\mathcal{V}_0| = 8$ vertices, arbitrarily ordered. For a pooling of size 4, two coarsenings of size 2 are needed: let Graclus gives $\mathcal{G}_1$ of size $n_1 = |\mathcal{V}_1| = 5$, then $\mathcal{G}_2$ of size $n_2 = |\mathcal{V}_2| = 3$, the coarsest graph. Sizes are thus set to $n_2 = 3$, $n_1 = 6$, $n_0 = 12$ and fake nodes (in blue) are added to $\mathcal{V}_1$ (1 node) and $\mathcal{V}_0$ (4 nodes) to pair with the singeltons (in orange), such that each node has exactly two children. Nodes in $\mathcal{V}_2$ are then arbitrarily ordered and nodes in $\mathcal{V}_1$ and $\mathcal{V}_0$ are ordered consequently. At that point the arrangement of vertices in $\mathcal{V}_0$ permits a regular 1D pooling on $x \in \mathbb{R}^{12}$ such that $z = [\max(x_0, x_1), \max(x_4, x_5, x_6), \max(x_8, x_9, x_{10})] \in \mathbb{R}^3$, where the signal components $x_2, x_3, x_7, x_{11}$ are set to a neutral value.

analog to pooling a regular 1D signal. Figure 2 shows an example of the whole process. This regular arrangement makes the operation very efficient and satisfies parallel architectures such as GPUs as memory accesses are local, i.e. matched nodes do not have to be fetched.

## 3 Related Works

### 3.1 Graph Signal Processing

The emerging field of GSP aims at bridging the gap between signal processing and spectral graph theory [6, 3, 21], a blend between graph theory and harmonic analysis. A goal is to generalize fundamental analysis operations for signals from regular grids to irregular structures embodied by graphs. We refer the reader to [31] for an introduction of the field. Standard operations on grids such as convolution, translation, filtering, dilatation, modulation or downsampling do not extend directly to graphs and thus require new mathematical definitions while keeping the original intuitive concepts. In this context, the authors of [12, 8, 10] revisited the construction of wavelet operators on graphs and techniques to perform mutli-scale pyramid transforms on graphs were proposed in [32, 27]. The works of [34, 25, 26] redefined uncertainty principles on graphs and showed that while intuitive concepts may be lost, enhanced localization principles can be derived.

### 3.2 CNNs on Non-Euclidean Domains

The Graph Neural Network framework [29], simplified in [20], was designed to embed each node in an Euclidean space with a RNN and use those embeddings as features for classification or regression of nodes or graphs. By setting their *transition function* $f$ as a simple diffusion instead of a neural net with a recursive relation, their *state* vector becomes $s = f(x) = Wx$. Their point-wise *output function* $g_\theta$ can further be set as $\hat{x} = g_\theta(s, x) = \theta(s - Dx) + x = \theta Lx + x$ instead of another neural net. The Chebyshev polynomials of degree $K$ can then be obtained with a $K$-layer GNN, to be followed by a non-linear layer and a graph pooling operation. Our model can thus be interpreted as multiple layers of diffusions and node-local operations.

The works of [11, 7] introduced the concept of constructing a local receptive field to reduce the number of learned parameters. The idea is to group together features based upon a measure of similarity such as to select a limited number of connections between two successive layers. While this model reduces the number of parameters by exploiting the locality assumption, it did not attempt to exploit any stationarity property, i.e. no weight-sharing strategy. The authors of [4] used this idea for their spatial formulation of graph CNNs. They use a weighted graph to define the local neighborhood and compute a multiscale clustering of the graph for the pooling operation. Inducing weight sharing in a spatial construction is however challenging, as it requires to select and order the neighborhoods when a problem-specific ordering (spatial, temporal, or otherwise) is missing.

A spatial generalization of CNNs to 3D-meshes, a class of smooth low-dimensional non-Euclidean spaces, was proposed in [23]. The authors used geodesic polar coordinates to define the convolu-

| Model | Architecture | Accuracy |
|---|---|---|
| Classical CNN | C32-P4-C64-P4-FC512 | 99.33 |
| Proposed graph CNN | GC32-P4-GC64-P4-FC512 | 99.14 |

Table 1: Classification accuracies of the proposed graph CNN and a classical CNN on MNIST.

tion on mesh patches, and formulated a deep learning architecture which allows comparison across different manifolds. They obtained state-of-the-art results for 3D shape recognition.

The first spectral formulation of a graph CNN, proposed in [4], defines a filter as

$$g_\theta(\Lambda) = B\theta, \tag{7}$$

where $B \in \mathbb{R}^{n \times K}$ is the cubic B-spline basis and the parameter $\theta \in \mathbb{R}^K$ is a vector of control points. They later proposed a strategy to learn the graph structure from the data and applied the model to image recognition, text categorization and bioinformatics [13]. This approach does however not scale up due to the necessary multiplications by the graph Fourier basis $U$. Despite the cost of computing this matrix, which requires an EVD on the graph Laplacian, the dominant cost is the need to multiply the data by this matrix twice (forward and inverse Fourier transforms) at a cost of $\mathcal{O}(n^2)$ operations per forward and backward pass, a computational bottleneck already identified by the authors. Besides, as they rely on smoothness in the Fourier domain, via the spline parametrization, to bring localization in the vertex domain, their model does not provide a precise control over the local support of their kernels, which is essential to learn localized filters. Our technique leverages on this work, and we showed how to overcome these limitations and beyond.

## 4 Numerical Experiments

In the sequel, we refer to the non-parametric and non-localized filters (2) as *Non-Param*, the filters (7) proposed in [4] as *Spline* and the proposed filters (4) as *Chebyshev*. We always use the Graclus coarsening algorithm introduced in Section 2.2 rather than the simple agglomerative method of [4]. Our motivation is to compare the learned filters, not the coarsening algorithms.

We use the following notation when describing network architectures: FC$k$ denotes a fully connected layer with $k$ hidden units, P$k$ denotes a (graph or classical) pooling layer of size and stride $k$, GC$k$ and C$k$ denote a (graph) convolutional layer with $k$ feature maps. All FC$k$, C$k$ and GC$k$ layers are followed by a ReLU activation $\max(x, 0)$. The final layer is always a softmax regression and the loss energy $E$ is the cross-entropy with an $\ell_2$ regularization on the weights of all FC$k$ layers. Mini-batches are of size $S = 100$.

### 4.1 Revisiting Classical CNNs on MNIST

To validate our model, we applied it to the Euclidean case on the benchmark MNIST classification problem [19], a dataset of 70,000 digits represented on a 2D grid of size $28 \times 28$. For our graph model, we construct an 8-NN graph of the 2D grid which produces a graph of $n = |\mathcal{V}| = 976$ nodes ($28^2 = 784$ pixels and 192 fake nodes as explained in Section 2.3) and $|\mathcal{E}| = 3198$ edges. Following standard practice, the weights of a $k$-NN similarity graph (between features) are computed as

$$W_{ij} = \exp\left(-\frac{\|z_i - z_j\|_2^2}{\sigma^2}\right), \tag{8}$$

where $z_i$ is the 2D coordinate of pixel $i$.

This is an important sanity check for our model, which must be able to extract features on any graph, including the regular 2D grid. Table 1 shows the ability of our model to achieve a performance very close to a classical CNN with the same architecture. The gap in performance may be explained by the isotropic nature of the spectral filters, i.e. the fact that edges in a general graph do not possess an orientation (like up, down, right and left for pixels on a 2D grid). Whether this is a limitation or an advantage depends on the problem and should be verified, as for any invariance. Moreover, rotational invariance has been sought: (i) many data augmentation schemes have used rotated versions of images and (ii) models have been developed to learn this invariance, like the Spatial Transformer Networks [14]. Other explanations are the lack of experience on architecture design and the need to investigate better suited optimization or initialization strategies.

The LeNet-5-like network architecture and the following hyper-parameters are borrowed from the TensorFlow MNIST tutorial[2]: dropout probability of 0.5, regularization weight of $5 \times 10^{-4}$, initial

| Model | Accuracy |
|---|---|
| Linear SVM | 65.90 |
| Multinomial Naive Bayes | 68.51 |
| Softmax | 66.28 |
| FC2500 | 64.64 |
| FC2500-FC500 | 65.76 |
| GC32 | 68.26 |

Table 2: Accuracies of the proposed graph CNN and other methods on 20NEWS.

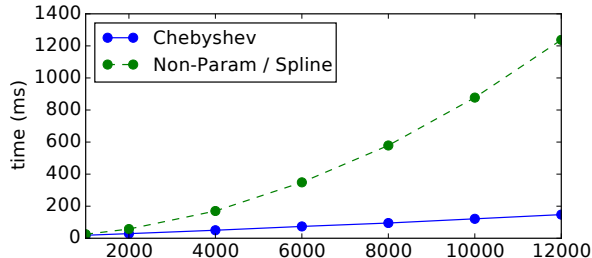

Figure 3: Time to process a mini-batch of $S = 100$ 20NEWS documents w.r.t. the number of words $n$.

| Dataset | Architecture | Accuracy | | |
|---|---|---|---|---|
| | | Non-Param (2) | Spline (7) [4] | Chebyshev (4) |
| MNIST | GC10 | 95.75 | 97.26 | 97.48 |
| MNIST | GC32-P4-GC64-P4-FC512 | 96.28 | 97.15 | 99.14 |

Table 3: Classification accuracies for different types of spectral filters ($K = 25$).

| Model | Architecture | Time (ms) | | Speedup |
|---|---|---|---|---|
| | | CPU | GPU | |
| Classical CNN | C32-P4-C64-P4-FC512 | 210 | 31 | 6.77x |
| Proposed graph CNN | GC32-P4-GC64-P4-FC512 | 1600 | 200 | 8.00x |

Table 4: Time to process a mini-batch of $S = 100$ MNIST images.

learning rate of 0.03, learning rate decay of 0.95, momentum of 0.9. Filters are of size $5 \times 5$ and graph filters have the same support of $K = 25$. All models were trained for 20 epochs.

## 4.2 Text Categorization on 20NEWS

To demonstrate the versatility of our model to work with graphs generated from unstructured data, we applied our technique to the text categorization problem on the 20NEWS dataset which consists of 18,846 (11,314 for training and 7,532 for testing) text documents associated with 20 classes [15]. We extracted the 10,000 most common words from the 93,953 unique words in this corpus. Each document $x$ is represented using the bag-of-words model, normalized across words. To test our model, we constructed a 16-NN graph with (8) where $z_i$ is the word2vec embedding [24] of word $i$, which produced a graph of $n = |\mathcal{V}| = 10,000$ nodes and $|\mathcal{E}| = 132,834$ edges. All models were trained for 20 epochs by the Adam optimizer [17] with an initial learning rate of 0.001. The architecture is GC32 with support $K = 5$. Table 2 shows decent performances: while the proposed model does not outperform the multinomial naive Bayes classifier on this small dataset, it does defeat fully connected networks, which require much more parameters.

## 4.3 Comparison between Spectral Filters and Computational Efficiency

Table 3 reports that the proposed parametrization (4) outperforms (7) from [4] as well as non-parametric filters (2) which are not localized and require $\mathcal{O}(n)$ parameters. Moreover, Figure 4 gives a sense of how the validation accuracy and the loss $E$ converges w.r.t. the filter definitions.

Figure 3 validates the low computational complexity of our model which scales as $\mathcal{O}(n)$ while [4] scales as $\mathcal{O}(n^2)$. The measured runtime is the total training time divided by the number of gradient steps. Table 4 shows a similar speedup as classical CNNs when moving to GPUs. This exemplifies the parallelization opportunity offered by our model, who relies solely on matrix multiplications. Those are efficiently implemented by cuBLAS, the linear algebra routines provided by NVIDIA.

## 4.4 Influence of Graph Quality

For any graph CNN to be successful, the statistical assumptions of locality, stationarity, and compositionality regarding the data must be fulfilled on the graph where the data resides. Therefore, the learned filters' quality and thus the classification performance critically depends on the quality of

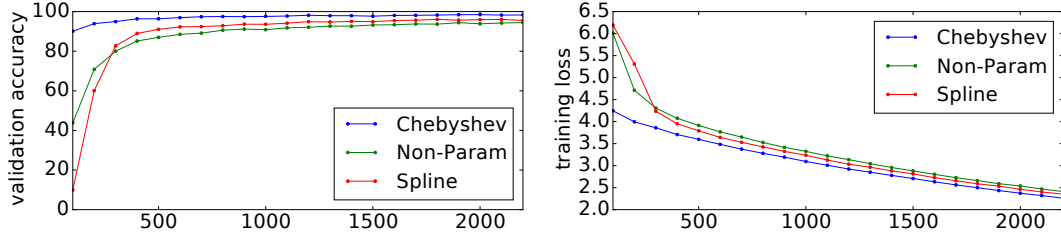

Figure 4: Plots of validation accuracy and training loss for the first 2000 iterations on MNIST.

| Architecture | 8-NN on 2D Euclidean grid | random |
|---|---|---|
| GC32 | 97.40 | 96.88 |
| GC32-P4-GC64-P4-FC512 | 99.14 | 95.39 |

Table 5: Classification accuracies with different graph constructions on MNIST.

| | word2vec | | | |
|---|---|---|---|---|
| bag-of-words | pre-learned | learned | approximate | random |
| 67.50 | 66.98 | 68.26 | 67.86 | 67.75 |

Table 6: Classification accuracies of GC32 with different graph constructions on 20NEWS.

the graph. For data lying on Euclidean space, experiments in Section 4.1 show that a simple $k$-NN graph of the grid is good enough to recover almost exactly the performance of standard CNNs. We also noticed that the value of $k$ does not have a strong influence on the results. We can witness the importance of a graph satisfying the data assumptions by comparing its performance with a random graph. Table 5 reports a large drop of accuracy when using a random graph, that is when the data structure is lost and the convolutional layers are not useful anymore to extract meaningful features.

While images can be structured by a grid graph, a feature graph has to be built for text documents represented as bag-of-words. We investigate here three ways to represent a word $z$: the simplest option is to represent each word as its corresponding column in the bag-of-words matrix while, another approach is to learn an embedding for each word with word2vec [24] or to use the pre-learned embeddings provided by the authors. For larger datasets, an approximate nearest neighbors algorithm may be required, which is the reason we tried LSHForest [2] on the learned word2vec embeddings. Table 6 reports classification results which highlight the importance of a well constructed graph.

## 5 Conclusion and Future Work

In this paper, we have introduced the mathematical and computational foundations of an efficient generalization of CNNs to graphs using tools from GSP. Experiments have shown the ability of the model to extract local and stationary features through graph convolutional layers. Compared with the first work on spectral graph CNNs introduced in [4], our model provides a strict control over the local support of filters, is computationally more efficient by avoiding an explicit use of the Graph Fourier basis, and experimentally shows a better test accuracy. Besides, we addressed the three concerns raised by [13]: (i) we introduced a model whose computational complexity is linear with the dimensionality of the data, (ii) we confirmed that the quality of the input graph is of paramount importance, (iii) we showed that the statistical assumptions of local stationarity and compositionality made by the model are verified for text documents as long as the graph is well constructed.

Future works will investigate two directions. On one hand, we will enhance the proposed framework with newly developed tools in GSP. On the other hand, we will explore applications of this generic model to important fields where the data naturally lies on graphs, which may then incorporate external information about the structure of the data rather than artificially created graphs which quality may vary as seen in the experiments. Another natural and future approach, pioneered in [13], would be to alternate the learning of the CNN parameters and the graph.

## Footnotes

[1] https://github.com/mdeff/cnn_graph

[2]https://www.tensorflow.org/versions/r0.8/tutorials/mnist/pros

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
