[Reviews · NeurIPS 2016]

Reviewer 1

Summary

This paper presents an extension of Convolutional Neural Networks to irregular grids in arbitrary dimension, using the properties of the Graph Laplacian Spectrum. The proposed architecture overcomes a computational burden of previous spectral methods, leading to efficient training and evaluation, as demonstrated on image classification and text categorization.

Qualitative Assessment

This paper builds upon previous work to propose a computationally efficient model that extends convolutions to irregular graphs. The major innovation is to express graph filtering with a compactly supported kernels as linear combinations of order K polynomials of the graph Laplacian, thus overcoming the burden of computing the Laplacian eigendecomposition, which is needed in both forward and backward passes in the previous spectral construction. Moreover, the Chebyshev recurrence implies that order K filters can be computed with O(K E) operations, where E is the number of edges and controls the cost of computing L x (L = Laplacian operator on the graph). This appears to be a very clean, efficient parametrization that translates into efficient learning and evaluation and overcomes an important computational limitation. I have a couple of comments that I would like the authors to develop during the rebuttal phase: - A previous model that operates in general graphs is the so-called Graph Neural Network (GNN) [Scarselli et al.,'09]. The basic model can be seen as follows. Given a generic vector-valued function x on a graph G, x: G--> R^d, we compute its linear diffusion Wx(i) = \sum_{j neighbor of i} w_{i,j} x(j) and its normalized scaling Dx(i) = (sum_{j neighbor of i} w_{i,j}) x(i). Then a "GNN" layer transforms x into Phi(x): G-->R^{d'} via a generic point-wise operator combining x and Wx: Phi(x)(i) = Psi( Dx(i), Wx(i)), with Psi being a generic pointwise operator: Psi: R^d x R^d --> R^{d'}. For example, if Psi(x,y) = x - y we obtain the Graph Laplacian Phi(x) = L(x), and thus the order-K graph filtering can be realized with a K-layer GNN using linear pointwise operators Psi. The model which computes filters as linear combinations of powers of graph laplacians can thus be seen as a special case of the previous model, which interleaves linear layers Psi (responsible of computing the powers of the laplacian) with nonlinear layers. It would be interesting for the authors to comment on the pros/cons of their model with respect to the general GNN. - The numerical experiments in MNIST show good performance of the graph CNN, but also a gap with respect to the classic CNN architectures. One can thus ask whether this gap is simply due to model tuning or due to some intrinsic limitations of graph CNNs when used on regular, low-dimensional grids. The filters learnt by the proposed graph CNN are polynomials of the Graph Laplacian, and therefore are rotationally invariant, whereas in the grid the Euclidean embedding allows for directional filters, which bring a crucial advantage to capture geometric structures in images. Is this analysis correct? In other words, there seems to be a fundamental price to pay for giving up the knowledge of the grid, its translation invariance and the notion of orientation. Obviously this only concerns the examples where the grid is available. Overall, and despite my previous two comments, this paper brings an important improvement to the table, that can extend the use of data driven, end-to-end learning with excellent learning complexity to other datasets beyond images, speech and text. I hope the rebuttal addresses my previous comments though!

Confidence in this Review

3-Expert (read the paper in detail, know the area, quite certain of my opinion)


Reviewer 2

Summary

The authors propose an improvement over existing approaches for using CNNs on data that cannot be represented as a grid graph but as a graph nevertheless. The work is motivated with applications in social network analysis, gene data, etc. The claimed advances are (a) a more efficient method for applying convolutions in Fourier space (from O(n^2) to O(E)) and (b) a pooling method that can be parallelized using GPUs.

Qualitative Assessment

I have two main criticisms of the paper. First, I find the presentation of the ideas and the theory rather poor. The exposition is confusing and requires more precise definitions. What exactly are the technical contributions of the authors remains difficult to distinguish from existing results. That's especially true for section 4 where the authors write densely and mix a lot of terminology. My recommendation would be to clean the paper up and provide precise definitions and a story line that's a bit easier to follow. The paper is very inaccessible in its current form. The second criticism concerns the experiments. It is not obvious to me that the results are definite. For instance, I find it strange that using random graphs results only in a small performance drop. This is especially striking in the 20NEWS data set. It makes me wonder what the point of the approach is if the graph structure has such minimal impact on the results. (Contrary to what the authors claim.) I scored the usefulness down because of this. Are there any theoretical results in this area? For instance, can you say something about the performance of your model compared to [13]. Your method is faster, but what about the effectiveness? Is it the same just faster? Sorry if I missed this.

Confidence in this Review

1-Less confident (might not have understood significant parts)


Reviewer 3

Summary

This paper presents a more efficient version of the model from Henaff-Bruna-LeCun.

Qualitative Assessment

I found it extremely hard to understand this paper, to the point where I had to read the cited Henaff-Bruna-LeCun paper in order to understand what this paper is even about. As far as I can tell this paper is presenting an efficient way to implement the algorithm from Henaff-Bruna-LeCun, which seems like a worthwhile thing to do since the algorithm in that paper is very computationally expensive. I'm surprised there are no "naturally" graph structured data sets in the experiments. The abstract alludes to a few and it would have been nice to see these.

Confidence in this Review

1-Less confident (might not have understood significant parts)


Reviewer 4

Summary

This paper proposes a sophisticated approach to implementing convolutional filters on non-Euclidean domains, like graphs, based on exploiting deep properties of the Laplacian operator. The work is based on exploiting the insights in a number of different areas, including computational harmonic analysis, signal processing, numerical analysis and machine learning. A very efficient procedure for computing convolutional filters over graphs is proposed and some simple experiments are given to illustrate the approach.

Qualitative Assessment

This is a highly promising paper, as it shows how to efficiently generalize the convolutional operator from rectangular (2D) domains to those with irregular geometry (graphs, social networks etc.). The approach is mathematically sophisticated, building on recent insights in a number of different areas. The paper is well written given the technical nature of the material, and the potential for significant impact is high. The results are somewhat limited, and the improvement in Table 4 etc. is not huge. But Figure 2 shows the high efficiency in computing convolutional filters using sophisticated Chebyshev polynomial approximation.

Confidence in this Review

3-Expert (read the paper in detail, know the area, quite certain of my opinion)


Reviewer 5

Summary

This paper introduces a graph signal processing based convolutional neural network formulation on generalized graphs. Experimental results are presented on MNIST and 20NEWS datasets.

Qualitative Assessment

Generalized convolutional neurals nets, applied to arbitrary graph structure are a new and upcoming area with only a few recent works. Regardless of the immaturity of the field, the practical applications that will benefit from such approaches are significantly main stream such as social networks. This work is providing a fresh look with a new formulation, and I believe it will contribute to the field in terms of both advancing this area and spawning new discussions.

Confidence in this Review

2-Confident (read it all; understood it all reasonably well)


Reviewer 6

Summary

The paper present an extension of the popular Convolutional Neural Networks to graphs. What makes the extension of classical CNN to graphs not trivial is the convolution operation, which is well defined on regular grids. The authors proposed a way to extend the convolution operation of CNNs to graphs by using tools from Signal Processing on Graphs and extending the previous work of Henaff-Bruna-LeCun. The authors tested their architecture on the classical MNIST and on 20NEWS. On MNIST the results showed that the graph-CNN approaches classical CNN performance if the graph approximate the regular grid of an image. On 20NEWS the results showed that, in the case when the data does not allow to apply classical CNNs, the convolution step of the graph-CNN improves over a non-convolutional architecture.

Qualitative Assessment

In my opinion this is a good work. It pushes forward the previous work of Henaff-Bruna-LeCun on extending CNNs to graphs and overcomes some of the limitations of the previous work. In particular: 1) It does not require an eigendecomposition of the graph Laplacian, contrary to the previous approach by Henaff-Bruna-LeCun. Such eigendecomposition becomes a limitation when the data size increases, since its complexity is cubic in the number of nodes. 2) The complexity of learning the convolutional filters is reduced from quadratic to linear in the number of nodes by replacing spline interpolation in the frequency domain with Chebyshev polynomials and exploiting the properties of the latter. 3) The memory required is also reduced from quadratic to linear. The paper presents some novel contributions. The main limitation of the present work is the limitation to work with a single graph. With the proposed approach in fact it is not possible to compare different graphs or learn some parameters on one graph (e.g. Facebook users) and then transfer them to another graph (e.g. Twitter users). The second main limitation, in my opinion, resides in the pooling strategy since it requires a processing step where to coarsen the graph. What is slightly less satisfying in my opinion are the experiments. The authors tested their architecture on two datasets: the classical MNIST and 20NEWS. On MNIST they showed performances close to a LeNet5-like architecture only in the case where the graph is a regular grid (trivial graph structure), while the performances drops by ~3% when the graph structure moves away from a regular grid. Although the main intent of the authors here was probably to show that in case of regular grids their performance approach the classical CNNs one, I want to remark that both the performances are far from the state-of-the-art. On 20NEWS experiment it is not clear to me whether the authors compared their performances with state-of-the-art methods on this dataset or only with baselines. Moreover, the improvement given by the convolution operation in the graph-CNN does not seem to be huge. Overall, I recommend the acceptance of this work.

Confidence in this Review

2-Confident (read it all; understood it all reasonably well)